

# MultNAT HIV-1 Viral Load Assay: a simple, rapid and sensitive cross-primer amplification assay for detecting the HIV-1 virus in the field

Lili Huang[1], Zhongbao Zuo[1], Lanlan Hu[1], Jing Wu[1], Qingting Bu[2], Yueping Li[2], Xiuhui Li[2], Yanqiong Zhou[2] and Aifang Xu[1]

[1] Clinical Laboratory, Hangzhou Xixi Hospital, Hangzhou, Zhejiang Province, China
[2] Clinical Laboratory, Ustar Biotechnologies, Hangzhou, Zhejiang Province, China

## ABSTRACT

**Background.** A large number of new human immunodeficiency virus (HIV) patients are diagnosed worldwide every year. This study aimed to design a rapid, efficient, and convenient kit for detecting HIV-1 viral load.

**Methods.** The sensitivity and specificity of the kit were determined, and serum samples were collected from HIV-negative and HIV-positive individuals who underwent medical checkups at Xixi Hospital in Hangzhou City, Zhejiang Province, China, between June and August 2023 for testing. The presence of HIV-1 in the clinical samples was assessed *via* a diagnostic PCR system and a conventional fluorescent quantitative PCR technique in this study.

**Results.** HIV-1 RNA was measured with a 100 copies/mL sensitivity *via* the MultNAT HIV-1 Viral Load Assay. The accuracy of the assay was as follows: The assay showed near-perfect agreement with compare kit (kappa = 0.9945), positive compliance rate of 99.45% (182/183), negative compliance rate of 100% (183/183). The linear correlation coefficient between MultNAT HIV-1 Viral Load Assay and DaAn kit was 0.95, and the R2 was 0.90.

**Conclusion.** Due to its simplicity, rapid response time, and high sensitivity and specificity, the MultNAT HIV-1 Viral Load Assay is suitable for primary care and epidemiological screening, facilitating rapid and accurate HIV-1 detection.

## INTRODUCTION

People with human immunodeficiency virus (PWH) are prone to various opportunistic infections and central nervous system lesions (*Barnett et al., 2008*). Decreased number of cluster of differentiation 4 (CD4)-positive T lymphocytes is a feature of human immunodeficiency virus (HIV) infection. Since the first HIV patient identified in the United States in 1981, HIV has spread rapidly throughout the world (*UNAIDS, 2021*). According to the Joint United Nations Program on HIV/AIDS (UNAIDS), by the end of 2022, there were 39 million patients with HIV/AIDS in the world, 1.3 million new HIV

Corresponding author
Aifang Xu, xuaifangxxh@163.com

infections, and 630,000 deaths in 2022 (*WHO World Health Organization, 2020*). In China, as of the end of 2022, there were 1.223 million patients with HIV/AIDS, 107,000 new HIV infections, and 3.0 million deaths (*Han, 2023*). HIV seriously jeopardizes human health and has become a public health problem of common concern to all mankind (*Wang & Zhong, 2015*).

However, 1.5 million new infections are still reported worldwide each year (*UNAIDS, 2022*). UNAIDS 2020 "90-90-90" (*UNAIDS, 2014*) targets, the first of which 90% of people living with HIV. It is estimated that people with primary HIV infection are between 10 and 26 times more infectious than those with chronic HIV infection and account for 38% to 50% of all HIV transmissions (*Ambrosioni et al., 2014*; *Ambrosioni et al., 2012*). The globally prevalent HIV viruses can be categorized into two types on the basis of differences in genetic sequence: HIV-1 and HIV-2. HIV-1 is widely distributed throughout the world and is the major strain that is responsible for the AIDS pandemic, with almost 95% of global HIV infections being caused by the HIV-1 type, whereas HIV-2 is largely confined to the African region (*Hemelaar, 2012*; *Geretti, 2006*). Detection of primary human immunodeficiency virus-1 (HIV-1) infection is critical for achieving the "95-95-95" goal. However, global progress remains insufficient, with many regions failing to meet the earlier "90-90-90" interim targets (*UNAIDS, 2014*). Current data indicate that the second and third "95-95-95" benchmarks-90% of people receiving antiretroviral therapy (ART) and 90% achieving sustained virological suppression—are lagging behind global objectives (*UNAIDS, 2021*). This shortfall presents a major challenge for ending the AIDS pandemic by 2030 and reaching the more ambitious "95-95-95" targets by 2025 (*UNAIDS, 2021*; *United Nations, 2021*).

Highly effective ART is considered the most effective treatment for HIV/AIDS and can significantly reduce mortality and complication rates in HIV/AIDS patients. Effective ART provides sustained suppression of the viral load in plasma (below 50 copies per milliliter/mL), and patients' viral loads are monitored by the accurate measurement of HIV-1 ribonucleic acid (RNA) levels in response to treatment (*World Health Organization, 2013*; *World Health Organization, 2010*). The HIV-1 viral load has also been used as a follow-up indicator in HIV-1-infected patients with AIDS to determine the effectiveness of treatment and the risk of transmission; thus, measuring the HIV-1 viral load is very important.

HIV infection can be definitively diagnosed by several types of laboratory tests, such as antibody-capture enzyme-linked immunosorbent assay (ELISA), antigen-detection tests, serological neutralization tests, real-time RT–PCR tests, and viral isolation cultures (*Cornett & Kirn, 2013*). The PCR method has a higher sensitivity and quicker reaction compared with other methods. The PCR quantification method requires four different concentrations of standard samples for quantification, which is also time-consuming and expensive. Additionally, handling HIV-infected blood samples poses biosafety risks, necessitating stringent Biosafety Level 2 (BSL-2) or higher laboratory conditions. Therefore, there is an urgent need for a quantitative HIV-1 nucleic acid test kit that can produce rapid results, replace manual or automated instrument-based sample nucleic acid extraction and purification, avoid the occurrence of cross-contamination, eliminate the need for

professionals to operate and analyze the data, and quantitatively detect HIV-1 nucleic acids. At present, the existing domestic HIV-1 nucleic acid quantitative detection kit cannot solve these problems well. This study aimed to design a rapid, efficient, and convenient kit for detecting of HIV-1 viral load.

## MATERIALS & METHODS

### Study design

We prospectively collected and tested 366 clinical blood samples from Xixi Hospital in Hangzhou, of which 183 were negative and 183 were positive. HIV-positive participants were enrolled if their viral load exceeded 100 copies/mL per the DaAn reference test, and the enrolled negative patients were from the HIV-negative hospitalized patients and HIV-positive patients with negative DaAn loads, and the healthy medical examination population included 20 people. The HIV-1 nucleic acid quantitative detection kit (PCR-fluorescent probe method) purchased from DaAn Gene Co., Ltd. is CE-IVD/NMPA-approved and widely used in clinical practice in China (*Li et al., 2023*). Given its high accuracy, reliability, and cost-effectiveness, we selected the DaAn Gene kit for our study. The study was approved by the Scientific Research Subcommittee of the Ethics Committee of Xixi Hospital, Hangzhou, China, with the approval number Hangzhou Xixi Medical Lun-Quan 2023 Research No. 005. Written informed consent from patients was waived, as the samples used were leftover samples from routine clinical examinations.

### The MultNAT HIV-1 Viral Load Assay used in this research

The MultNAT HIV-1 Viral Load Assay (Ustar Biotechnology Co., Ltd., Hangzhou, China) includes HIV-1 automated tubes, HIV-1 RNA extracts, and HIV-1-QS quantitative internal standards. The HIV-1 automated test tube is divided into the lysis-nucleic acid binding bead zone, the magnetic bead-nucleic acid washing zone and the nucleic acid elution-nucleic acid amplification zone from top to bottom. The nucleic acid elution-nucleic acid amplification zone comprises reaction zone 1 and reaction zone 2. Reaction zone 1 is used for the detection of the LTR gene, and reaction zone 2 is used for the detection of the pol gene (Fig. 1A). The HIV-1 automated test tube must be loaded into the nucleic acid analyzer (Ustar Biotechnology Co., Ltd., Hangzhou, China) for detection (Fig. 1B).

### Development of the formula for quantification of HIV-1 RNA viral load

In order to determine the titer of HIV-1 RNA, we optimized the system of internal quantitative standard nucleic acid in the HIV-1-QS which had a same amplification efficiency with the target nucleic acid. Viral load determination is based on the assumptions that the target nucleic acid and the internal quantitative standard nucleic acid are amplified with the same efficiency and that equal amounts of amplicon copies of target nucleic acid and quantitative standard nucleic acid are amplified and detected at the same cycle threshold (Ct). Therefore, the $\Delta$Ct ($Ct_{QS}-Ct_{target}$) is linear to log (target concentration/QS concentration), where in "QS" stands for the internal quantitative standard nucleic acid. For instance, the viral load of HIV-1 RNA can then be calculated by using a polynomial
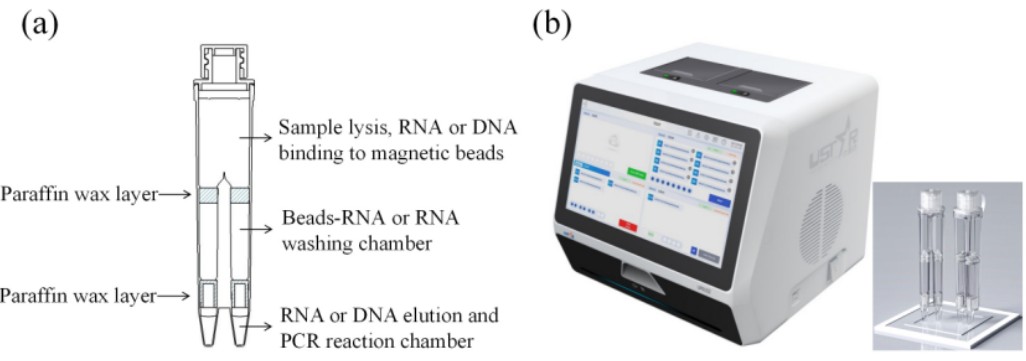

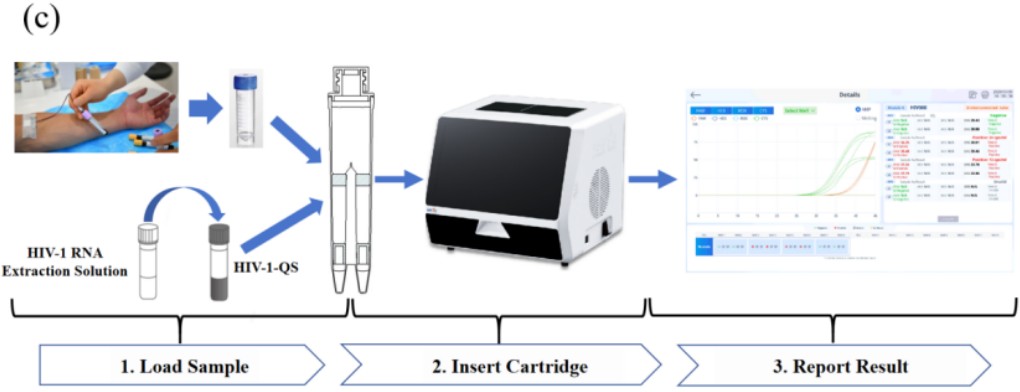

**Figure 1 Schematic diagram.** (A) Cutaway view of the cartridge (sketch map); (B) cutaway view of the equipment; (C) workflow of the MultNAT HIV-1 viral load test.

calibration formula as in the following equation:

$$\text{Viral load of HIV-1 RNA (cps/mL)} = 10^{(a(\Delta Ct)^2 + b(\Delta Ct) + c)} \tag{1}$$

$$\Delta Ct = Ct_{QS} - Ct_{target}. \tag{2}$$

According to the $\Delta Ct$ Eq. (2) of standard AcroMetrix™ HIV-1 Panel and HIV-1-QS, we determined the polynomial constants (a, b and c) in the Eq. (1). For a specific target, the polynomial constants (a, b and c) and the concentration of the quantitative standard nucleic acid are known, therefore the only variable in the equation is the difference $\Delta Ct$. The polynomial constants for the LTR target were $a = -0.0014$, $b = 0.2248$, $c = 4.2102$, and for the pol target were $a = 0.0195$, $b = 0.3969$, $c = 3.4798$. The fit formula had been built in the MultNAT instrument and the viral load of HIV-1 RNA can be interpreted automatically.

**Table 1  The primers and probe were designed to target the LTR gene and pol gene of HIV-1.** Show the relevant sequences of the LTR gene and pol gene of HIV-1.

| Item | Sequence (5′–3′) |
| --- | --- |
| HIV-1 LTR-F | GTAACTAGAGATCCCTCAGACC |
| HIV-1 LTR-R | GCGTCCTGAGCCGAACGAC |
| HIV-1 LTR-P | AAAATCTCTAGCAGTGGCGCCCG |
| HIV-1 pol-F | TCACAMATACAAACAACAGAA |
| HIV-1 pol-R | GTYGAGGACACCTTTCCACT |
| HIV-1 pol-P | AGAYCCTATYTGGAAAGGACC |

### Primers and probes used in the MultNAT HIV-1 Viral Load Assay

The MultNAT HIV-1 Viral Load Assay was designed with primer pairs and probes for the HIV-1 LTR and pol genes, and the primers were synthesized by Shanghai Sangong Biotechnology Co. (Shanghai, China) (Table 1 & Fig. 2). To assay the samples, extracts were chemically lysed at 94 °C in a nucleic acid analyzer (Ustar Biotechnology Co., Ltd., Hangzhou, China) to release nucleic acids. The nucleic acids from the samples that were used for the assay were permeated through the nucleic acid magnetic guide of the nucleic acid analyzer, passed through the different detection zones, purified in the magnetic bead–nucleic acid wash zone, and finally eluted at the bottom of the nucleic acid detection tube, where the amplification reaction occurred. With the use of the hot-start Taq enzyme and reverse transcriptase, the target sequences were amplified under isothermal conditions *via* amplification primers. Moreover, the fluorescent probe specifically bound to the target and generated a fluorescent signal. The fluorescence signal could be captured in real time during the nucleic acid amplification process and analyzed to automatically determine the test result and calculate the amount of HIV-1 RNA in the sample (in copies/mL) according to the built-in quantitative formula. The developed kit is a fully automated method for nucleic acid analysis, *i.e.,* lysis, binding, washing, elution, and amplification reactions are performed in a closed assay tube.

### Generation of a standard RNA template

The RNA template of HIV-1 LTR gene is from the HIV-1 RNA standard quantitative synthetic human immunodeficiency virus 1 (HIV-1) RNA (Cat: VR-3245SD) which is purchased from ATCC; the HIV-1 pol gene RNA template was transcribed from the pol plasmid template, the pol plasmid template was synthesized by Sangon Biotech (Shanghai, China) and the transcription reagents were purchased from Nanjing Novozymes Bioscience Co. (Nanjing, China).

### Sensitivity and specificity assessment of the MultNAT HIV-1 viral load assay

We used the AcroMetrix™ HIV-1 Panel from Thermo Fisher Scientific (Waltham, MA, USA) to assess the sensitivity of the Human Immunodeficiency Virus Type 1 Nucleic Acid Load Assay, and the concentration and quality of the extracted DNA were assessed *via* the MultNAT Molecular Diagnostic PCR System (Ustar Biotechnology Co., Ltd., Hangzhou, China). The concentration and quality of the extracted DNA were assessed.

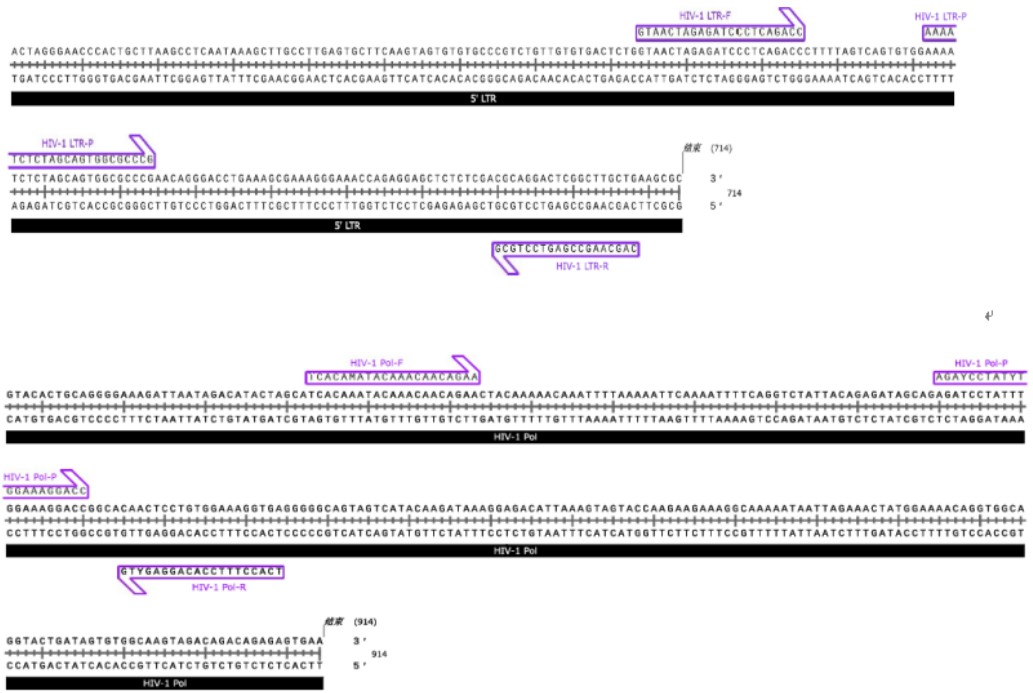

**Figure 2** **Primers and probes of HIV-1 LTR and pol genes.** Due to the high mutation frequency of HIV-1, we designed two sets of primer-probe systems to cover the detection of HIV-1 groups M, N, and O. Set 1 targets the LTR gene, including the primer pair HIV-1 LTR-F and HIV-1 LTR-R and the probe HIV-1 LTR-P, for the detection of HIV-1 groups M and N. Set 2 targets the Pol gene, including the primer pair HIV-1 Pol-F and HIV-1 Pol-R and the probe HIV-1 Pol-P, for the detection of HIV-1 group O.

MS2 phage broth (BeNa Culture Collection) was used as the negative serum substrate, and the AcroMetrix™ HIV-1 Panel was diluted with the negative serum substrate to obtain test concentrations of 1,000, 500, 100, and 50 copies/mL. Then, these samples were added to assay tubes and assayed on a fully automated medical PCR analyzer system with three replicates for each concentration. The lowest diluted concentration that was positive in all three replicates was defined as the limit of detection.

To further determine the specificity of the system, we utilized this kit to validate common clinical pathogens (human immunodeficiency virus type 2, human T-cellophilic viruses 1 and 2, *Candida albicans*, cytomegalovirus, human herpesvirus 4, hepatitis A virus, hepatitis B virus, hepatitis C virus, herpes simplex virus 1 and 2, human herpesvirus 6, influenza A virus, *Staphylococcus aureus*, *etc.*). The test was repeated three times for each common clinical pathogen.

## Clinical application of the MultNAT HIV-1 Viral Load Assay

From June to August 2023, HIV-negative and HIV-positive patients who underwent medical checkups at Xixi Hospital in Hangzhou City, Zhejiang Province, China, were prospectively enrolled in this study. The blood samples collected in this study were provided by the Laboratory Department of Xixi Hospital in Hangzhou City, China. The

obtained blood samples were centrifuged at 3,000 rpm to obtain serum, which was evenly divided into two portions.

### Fluorescence quantitative PCR

A portion of the collected serum or plasma samples was used as a control for fluorescence quantitative PCR with the Human Immunodeficiency Virus Type 1 (HIV-1) Nucleic Acid Quantitative Detection Kit (PCR-Fluorescent Probe Method) from DaAn Genetics Co. Fifteen microliters of HIV-1 reaction solution A and 5 μl of HIV-1 reaction solution B were mixed well, and after a short centrifugation to centrifuge all the liquid to the bottom of the tube, 4 μl of the internal standard solution and 40 μl of the sample were added, the mixture was centrifuged for a few seconds at 8,000 rpm, and then the mixture was transferred to the amplification detection zone. The procedure was carried out in strict accordance with the kit instructions. Finally, the test was performed *via* the MultNAT Molecular Diagnostic PCR System (Ustar Biotechnology Co., Ltd., Hangzhou, China).

### Kit assay

The HIV-1 RNA extract was added to the HIV-1-QS quantitative internal standard and fully dissolved to obtain the mixture. The mixture was added to the HIV-1-automatic detection tube, 400 μL of serum or plasma sample was added for testing, and PCR amplification was performed. The fluorescent probe specifically bound to the target and generated fluorescent signals, and the fluorescent signals were captured in real time *via* the MultNAT Molecular Diagnostic PCR System (Ustar Biotechnology Co., Ltd., Hangzhou, China), after which the test results were automatically determined by analyzing the changes in the fluorescent signals. The amount of HIV-1 RNA in the sample was calculated automatically according to the built-in quantitative formula (unit: copies/mL (Fig. 1C).

## Statistical analyses

The differences between the detection rates of the two methods were tested *via* McNemar's test. A *P* value of < 0.05 was considered statistically significant.

## RESULTS

### Sensitivity of the MultNAT HIV-1 viral load assay

We diluted the purchased AcroMetrix™ HIV-1 Panel to the appropriate concentration and measured the sensitivity of the Human Immunodeficiency Virus Type 1 Viral Load Assay for detecting HIV-1 RNA to 100 copies/ml (Table 2).

### Specificity of the MultNAT HIV-1 Viral Load Assay Kit

Our analysis of the 14 common clinical pathogens revealed no positive results, indicating that the MultNAT HIV-1 Viral Load Assay Kit (Ustar Biotechnology Co., Ltd., Hangzhou, China) has no cross-reactivity with the 14 common clinical pathogens (Table 3).

### Clinical performance of the MultNAT HIV-1 Viral Load Assay Kit

A total of 366 clinical samples were tested, 183 of them were negative, and 183 were positive based on the DaAn results (Table 4). Among 366 samples, only one patient with confirmed

**Table 2  Detection limit of the MultNAT HIV-1 viral load assay.** Sensitivity of the MultNAT HIV-1 viral load assay for detecting HIV-1 RNA to 100 copies/ml.

| Copies/number of DNA molecules | Positive rate |
|---|---|
| 50 | 2/3 |
| 100 | 3/3 |
| 500 | 3/3 |
| 1,000 | 3/3 |
| NTC (MS2) | 0 |

**Table 3  Specifcity of the MultNAT HIV-1 viral load assay.** Our assay has no cross-reactivity with the 14 common clinical pathogens.

| Pathogen | Positive rate |
|---|---|
| Immunodeficiency Virus type 2 | 0/3 |
| Human T-cell virus type 1 | 0/3 |
| Human T-cell virus type 2 | 0/3 |
| Candida albicans | 0/3 |
| Cytomegalovirus (CMV) | 0/3 |
| Human herpesvirus 4 (HSV) | 0/3 |
| Hepatitis A virus | 0/3 |
| Hepatitis B virus | 0/3 |
| Hepatitis C virus | 0/3 |
| Herpes simplex virus types 1 | 0/3 |
| Herpes simplex virus types 2 | 0/3 |
| Human herpesvirus 6 | 0/3 |
| Influenza A virus | 0/3 |
| Staphylococcus aureus | 0/3 |

**Table 4  Detection results of clinical samples of MultNAT HIV-1 viral load assay.** The accuracy of the MultNAT Molecular Diagnostic PCR System and its consistency with clinical diagnosis were as follows: kappa value of 0.9945; positive compliance rate of 99.45% (95% CI [98.03%–99.93%]); negative compliance rate of 100% (95% CI [97.99%–100.00%]); and total compliance rate of 99. 73% (95% CI [97.99%–100.00%]).

| Results of the kit of the present invention | Comparison of kit results | | Amount |
|---|---|---|---|
| | Positive | Negatives | |
| Positive | 182 | 0 | 182 |
| Negatives | 1 | 183 | 184 |
| Amount | 183 | 183 | 366 |
| Positive compliance rate | | 99.45% | |
| Negative compliance rate | | 100.00% | |
| Overall compliance rate | | 99.73% | |
| Kappa value | | 0.9945 | |

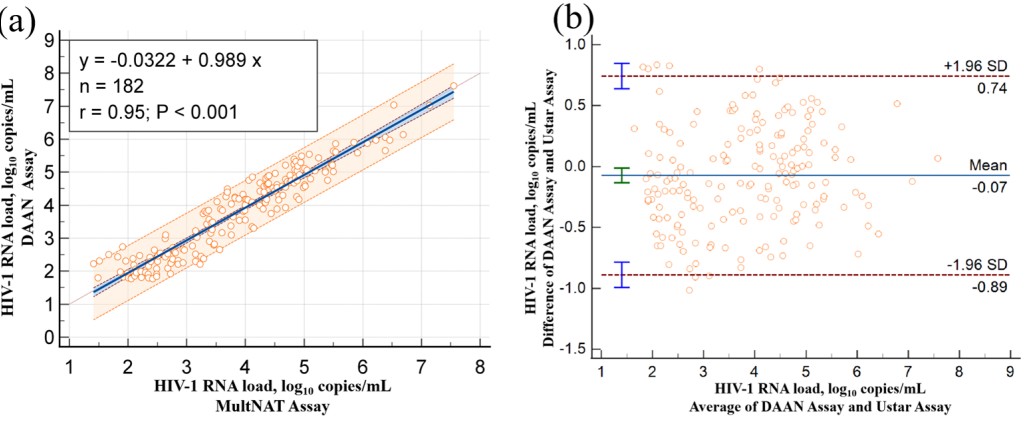

**Figure 3 Compare of Ustar MultNAT HIV-1 viral load assay and DaAN assay.** Passing-Bablok regression plot (A) and Bland-Altman plot (B) comparing Ustar MultNAT HIV-1 viral load assay against the DaAN assay.

positive HIV viral load was diagnosed negative by MultNAT HIV-1 Viral Load Assay, but all the others were fully matched.

The accuracy of the MultNAT Molecular Diagnostic PCR System and its consistency with clinical diagnosis were as follows: kappa value of 0.9945; positive compliance rate of 99.45% (95% CI [98.03%–99.93%]); negative compliance rate of 100% (95% CI [97.99%–100.00%]); and total compliance rate of 99.73% (95% CI [97.99%–100.00%]). The positive compliance rate, negative compliance rate and total compliance rate were all ≥95%. The linear correlation coefficient (r) between the kit used in the present study and the comparison kit was 0.95, the R2 was 0.90, and the correlation coefficient was $y = 0.989x - 0.0322$ (Table 4 & Fig. 3), which proved that the correlation coefficient between the kit used in the present study and the comparison kit was good.

## DISCUSSION

With the number of AIDS patients increasing daily, the early detection, diagnosis, and treatment of AIDS are among the most effective means to slow the spread of HIV and control the spread of people living with HIV (PLHIV). Here we develop a MultNAT HIV-1 Viral Load Assay for HIV-1 detection that is highly sensitive, simple, and quick to use. Moreover, this assay has potential for wide clinical application in the future, especially in low-developed countries.

There were four advantages compared to previous products (*Li et al., 2023*; *Zhao et al., 2021*). First, we used serum or plasma to detect the viral load, while other methods require RNA/DNA extraction, which can save time and prevent cross-contamination in our test. Second, the assay demonstrates exceptional efficiency, with 102 min from sample collection to result delivery. Although there was no specific time in previous research, depending on their method and description, we concluded the assay takes the least amount of time to achieve results. Third, our system can directly display qualitative or quantitative results of viral load on the instrument screen, and even non-medical workers can complete

the experiment and read the results within 5 min of training. The ease of operation of our reagent is evident compared to other reagents. Fourth, our assay maintains high sensitivity, with LOD of 100 copies/mL and 116 copies/mL for clinical samples. This unique combination of speed, accuracy, and operational simplicity effectively bridges existing gaps in HIV monitoring, especially in resource-limited regions with constrained laboratory infrastructure (*Drain et al., 2019*; *World Health Organization, 2022*).

At the 200 copies/mL threshold, our MultNAT HIV-1 Viral Load Assay's rapidity allows clinicians to promptly adjust treatment plan, potentially curbing resistance development. While other rapid tests (*Niemz, Ferguson & Boyle, 2011*; *Drain et al., 2019*; *Burchard et al., 2014*) achieve similar detection limits, their longer processing times and complex protocols limit clinical utility. Our optimized chemistry and streamlined workflow enable same-day clinical decisions, supporting the UNAIDS 95-95-95 targets by reducing the "detection-to-action" gap (*UNAIDS, 2021*).

According to the "China AIDS Diagnosis and Treatment Guidelines (2021 Edition)" (*Hepatitis C AIDS Group of the Infectious Diseases Branch of the Chinese Medical Association & Chinese Center for Disease Control and Prevention, 2021*) and the "U.S. Department of Health and Human Services Antiretroviral Therapy Guidelines (2018 Edition)", a viral load of 200 copies/ml is used as an indicator of the frequency of viral testing and the need to adjust the antiretroviral therapy regimen (*Panel on Antiretroviral Guidelines for Adults and Adolescents, 2018*), and it has been noted that the quantification of HIV nucleic acid is very important for monitoring disease and ART (*Zhao et al., 2021*) and that HIV test reagents have high clinical sensitivity (*World Health Organization, 2022*; *Burchard et al., 2014*; *Sollis et al., 2014*; *Drain et al., 2019*). The quantification of HIV nucleic acid levels is important for disease monitoring and ART (*Mellors et al., 1996*), the level of HIV is closely related to the development of the disease and the prognosis of HIV infection, and the quantification of the HIV viral load in the plasma of infected patients has important clinical value in predicting the course of the disease, formulating a treatment plan, monitoring the effectiveness of medication, and assisting in diagnosis. In this context, we designed a single modular HIV-RNA assay kit, which is simple to use, facilitates clinical testing and real-time monitoring, is easy to disseminate in large quantities because of its fast results and low cost of consumables for early HIV screening, and is methodologically superior to traditional assays such as western blotting.

Therefore, the highly sensitive MultNAT HIV-1 Viral Load Assay designed in this study can be used for the early diagnosis of HIV-1 infection, auxiliary diagnosis of difficult samples and monitoring of AIDS. By diluting the purchased AcroMetrix™ HIV-1 Panel to the appropriate concentration, we measured the sensitivity of the Human Immunodeficiency Virus Type 1 Nucleic Acid Load Assay to detect HIV-1 RNA at 100 copies/mL, which proves the high sensitivity of the assay. Moreover, we analyzed and tested 14 common clinical pathogens and did not obtain positive results, which confirmed the high specificity of the MultNAT HIV-1 Viral Load Assay (Ustar Biotechnology Co., Ltd., Hangzhou, China) and demonstrated that it has no cross-reactivity with the 14 common clinical pathogens.

To demonstrate its clinical performance, we compared the results of the same sample with the results of the MultNAT HIV-1 Viral Load Assay (PCR-fluorescent probe method)

from DaAn Genetics Co. The total positive compliance rate was 99.73% (out of 366 clinical samples). The degree of agreement between the accuracy of the MultNAT Molecular Diagnostic PCR System and the clinical diagnosis was as follows: kappa value, 0.9945; positive compliance rate, 99.45% (95% CI [98.03%–99.93%]); and negative compliance rate, 100% (95% CI [97.99%–100.00%]). The positive compliance rate, negative compliance rate, and total compliance rate were all ≥95%. The linear correlation ($r$) between the kit of the present invention and the comparison kit was 0.95, and the $R^2$ was 0.90, demonstrating a good consistency between the kit of the DaAn kit (using four different concentrations of standard samples for quantification) and the MultNAT HIV-1 Viral Load Assay. Although the assay exhibits significant advantages, the current HIV RNA detection kit has several limitations that warrant discussion. First, while demonstrating excellent sensitivity for LTR and pol gene targets, the assay's diagnostic coverage is incomplete as it lacks gag gene detection capability—a feature routinely incorporated in commercial triple-target assays. This single-gene omission may potentially compromise the assay's ability to detect viral mutations. Second, due to budgetary limitations, the comparison reagent we use is DaAn kit instead of the Roche HIV RT-PCR assay (a WHO-approved gold standard), which may affect the further exploitation of the results. Third, while preliminary results are promising, comprehensive blinded clinical trials with larger, genetically diverse patient cohorts will be essential to fully validate the assay's performance across different HIV subtypes and clinical settings.

## CONCLUSIONS

The MultNAT HIV-1 Viral Load Assay developed in this study is highly sensitive, simple, economical, and portable for primary care and epidemiologic screening. With its advantages in cost-effectiveness and operational simplicity, this assay holds significant potential not only for widespread clinical application in China, but also for improving HIV testing accessibility in resource-limited settings globally.

### Funding
This work was supported by the Hangzhou biomedical and health industry development support science and technology project (2022WJC284). The funders had no role in study design, data collection and analysis, decision to publish, or preparation of the manuscript.

### Grant Disclosures
The following grant information was disclosed by the authors:
Hangzhou Biomedical and Health Industry Development Support Science and Technology: 2022WJC284.

### Competing Interests
Qingting Bu, Yueping Li, Xiuhui Li and Yanqiong Zhou are employed by Ustar Biotechnologies

## Author Contributions

- Lili Huang conceived and designed the experiments, performed the experiments, analyzed the data, prepared figures and/or tables, authored or reviewed drafts of the article, and approved the final draft.
- Zhongbao Zuo conceived and designed the experiments, performed the experiments, analyzed the data, prepared figures and/or tables, authored or reviewed drafts of the article, and approved the final draft.
- Lanlan Hu conceived and designed the experiments, performed the experiments, authored or reviewed drafts of the article, and approved the final draft.
- Jing Wu conceived and designed the experiments, performed the experiments, authored or reviewed drafts of the article, and approved the final draft.
- Qingting Bu conceived and designed the experiments, performed the experiments, analyzed the data, prepared figures and/or tables, authored or reviewed drafts of the article, and approved the final draft.
- Yueping Li conceived and designed the experiments, performed the experiments, prepared figures and/or tables, and approved the final draft.
- Xiuhui Li conceived and designed the experiments, performed the experiments, prepared figures and/or tables, and approved the final draft.
- Yanqiong Zhou conceived and designed the experiments, performed the experiments, authored or reviewed drafts of the article, and approved the final draft.
- Aifang Xu conceived and designed the experiments, performed the experiments, authored or reviewed drafts of the article, and approved the final draft.

## Human Ethics

The following information was supplied relating to ethical approvals (*i.e.*, approving body and any reference numbers):

The study was approved by the Scientific Research Subcommittee of the Ethics Committee of Xixi Hospital, Hangzhou, China, with the approval number Hangzhou Xixi Medical Lun-Quan 2023 Research No. 005.

## Data Availability

The raw measurements are available in the Supplementary Files 1.

## Supplemental Information

Supplemental information for this article can be found online at http://dx.doi.org/10.7717/peerj.19753#supplemental-information.

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
