# Peer review of "MultNAT HIV-1 Viral Load Assay: a simple, rapid and sensitive cross-primer amplification assay for detecting the HIV-1 virus in the field"

_PeerJ, doi:10.7717/peerj.19753_

## Round 0.1 · original submission · Major Revisions

After thoroughly reviewing your manuscript, we have concluded that it requires significant revisions before it can be considered for publication. While this study is relevant to the field of HIV-1 diagnosis, it warrants strong consideration for publication, provided that the concerns regarding the study's limitations and the need for additional validation are adequately addressed. Additionally, the quality of English in the manuscript needs improvement to ensure clarity for readers. Please take the reviewers' feedback and comments into careful consideration before submitting a revised version of your manuscript.

**Language Note:** The review process has identified that the English language must be improved. PeerJ can provide language editing services - please contact us at [email protected] for pricing (be sure to provide your manuscript number and title). Alternatively, you should make your own arrangements to improve the language quality and provide details in your response letter. – PeerJ Staff

Reviewer 1 ·

Basic reporting

The study described diagnostic procedures for HIV detection using the MultiNAT HIV-1 Viral Load Assay Kit. The results showed potential results compared to the Daan Viral load kit. Study findings were promising, and the MultiNAT HIV-1 VL assay kit performed equal to the Daan VL Kit. The LOD is 100 cp/mL, as mentioned by the author.

However, the authors needed to provide additional information in the manuscript.

Introduction
The author needs to improve the English language and references to support the argument.
Line 40-42: HIV is secondary to various opportunistic infection?? What does it mean?
Line 76: AIDS patient samples pose an extreme biohazard risk?? Working in BSL-2 is okay for HIV Samples. Why AIDS, not HIV?
Line 96: The author must address the publication using point-of-care testing or real-time PCR for HIV Viral load recommended by WHO. What is a domestic HIV quantitative detection kit?

Material and methods
The author did not mention the DAAN kit. Why did the author use this kit as the gold standard?
Line 90, it is 1000 rpm?

Discussion
The author's claim in line 53 needs references to be a comparison. Can LOD 100cp/mL be defined as highly sensitive?

Experimental design

The study finding is not new but may have the potential for an HIV viral load testing kit alternative. The sample size is good, using AcroMetrix# HIV-1 for specificity and sensitivity, and compared with the DAAN HIV VL kit. The justification for using the DAAN HIV VL kit was needed.:

Improvement
There is no primer-probe design described in this study

Validity of the findings

Please refer to the previous point mentioned for the justification of the gold standard kit used in this study.

Additional comments

no comments

Reviewer 2 ·

Basic reporting

Upon reviewing the article, I believe it aligns well with the basic reporting standards.

Experimental design

The methodology is generally well-structured; there is just a typo in line 190 (1000 rpm instead of 1000 r).

Validity of the findings

In terms of the validity of the findings, I believe the study presents well-supported results. The methodology appears robust, and the conclusions are logically drawn from the data. Overall, the approach used in this study strengthens the validity of the results, and I see no significant issues in this regard.

Additional comments

This study, conducted between June and August 2023, involved HIV-negative and HIV-positive patients undergoing medical checkups at Xixi Hospital in Hangzhou, China. The primary aim was to validate the clinical application of the MultNAT HIV-1 Viral Load Assay kit. The results demonstrated that the kit offers high sensitivity and specificity, making it particularly valuable for early HIV-1 diagnosis and viral load monitoring, with strong correlation to traditional methods.
The significance of this study lies in its presentation of a fast, sensitive, and cost-effective approach for detecting HIV-1 at an early stage and monitoring the effectiveness of antiretroviral treatment. This method can enhance screening practices in primary care settings and facilitate ongoing patient monitoring, ultimately supporting better management of HIV.

·

Basic reporting

Every thing is OK. Some points needs more clarity which I will mention below.

Experimental design

1. In Material and Method section, in study design, Please mention did you collected Naive patients or all positive patients were on ART?
2. How many low viral load and high viral load samples from HIV positive samples?

3. In comarison, you used Daan Gene. Please mention Is Daan Test is WHO prequalified HIV RT PCR diagnostic kit? If no then it can be not be taken as gold standard.

Validity of the findings

Good

Reviewer 4 ·

Basic reporting

This is a methodological study where a new kit for HIV-1 diagnosis is compared with another established method. The writing does not clearly explain and convey the authors' conclusions and demands improvement. Furthermore, the conclusions are not supported by the presented data and require further experiments to establish and validate their proposed method as a sufficient alternative for HIV-1 diagnosis.
English language should be improved to simplify the messages the authors want to convey, they need to be more direct and clear, it is hard to interpret form the writing how they concluded what they did.

Experimental design

ABSTRACT:
what is etc, improve English, and write to the point…

not clear what are the positive results of 182 of 366? how many were know as negative how many as positive.
I do not understand how the 99.45% and 100% were calculated outs op how many?
how many times the samples were tested? is it all on one cycle/run of the test? did you have samples that failed? these super high percentage are a bit suspicious.... nothing went wrong?

you say compared to : "conventional fluorescent quantitative PCR." did qPCR detected everything what was the lower threshold of CT (cycle threshold) of detection? what was the high? please provide and IQR ranges


INTRODUCTION:
the introduction is too general and not to the point of giving th background for the reasons and necessity of the study in hand... it has errors claims that are not supported by references (lines 93:95) , where have it been shown thar aerosol contain HIV-DNA or RNA??? the only thing that is reasonable is the high genetic diversity of viral populations (add reference)

line 54: 90-90-90 an then lines: 62 : 95-95-95 ? why the discrepancy, I think they refer to the 90 in line 62 and then in line 67 introduce the 95-95-95.

line 103: the method is time consuming ??? then why use it

4th Generation HIV-1 testing are detecting the p24 viral-antigen that its levels are detectable about 11 days post infection and can compensate for the windows period where human antibodies are not sufficiently detected, how does this kit deals with p24 detection?

METHODS:
what is DANN test? (line 111) give refence explain the test
line125 : which region in the pol? which viral protein?
please provide genomic regions of HXB2 refrence for the location of detection

line 136: it is not clear how he constants in the equion (a,b,c) were calulated for the LRT and POL

line 162: 162 The HIV-1 LTR gene RNA template was LTR RNA- what does that mean?

DISCOUSION:
line242: nowadays we use the term People with HIV (PWH) and not AIDS patients ..
AIDS is the final stage of HIV-1 infection

the conclusions drawing here is wrong.. what is the link between drug resistance and low-level viremia to this study? and the kit proposed for diagnostic???

Line 253: the authors discuss “monitoring the disease” which they probably refer to clinical checkups of PWH under ART, however the kit they proposed is for diagnostic not monitoring… how is this relevant to the study in hand, the discussion should discuss compared to other methods in the market

Line 280: I do not understand how a “better correlation” was reported. What was the Kappa for the comparator kit

Study limitations were not discussed… what are the draw backs here???

Validity of the findings

Nowadays, tests offer antigen and antibody by qPCR of at least 3 viral regions (LTR, GAG, POL) these 2 regions are not sufficient.
Still, I think this requires further validation of a golden standard like Determine HIV-1/2 Ag/Ab Combo for validation to validate infection?
When the test is performed the status of the sample should be unknow to the people performing the test..
In order to improve this study authors have to perform a blinded experiment of at least 100 samples were the scientist who preform the test are not aware to the sample status (HIV – or +) and that should be blinded and independently reported between the new proposed kit to the comparator methodology.
In the raw data table provided it is not clear what are all the NAs values? The rest of the samples did not have a result? Ot it means a negative result?

Additional comments

Figs:
Fig. 1 only panel A is relevant, panel B is a picture of an instrument that can be illustrated

Fig. 2 this fig can be merged with Fig.1 it is redundant, please use scientific appropriate software to produce illustrations (e.g., biorender)

Fig 3 is not understandable what is the Y axis ?? what are 50-100 cps/ml and the X axis is also HIV-1 RNA ? what does this mean? What do you want to show here???
What is MS2 ? it is not clearly exlined in the text
Fig. 4
What is panel B?? why there are minus values for HIV-1 RNA? What does is mean? , it seems both axis are HIV-1 RNA for two different tests, but the scales of the axis are completely different , how is this comparable?

The figure legend should be explanatory of the Figs , independent of the main text. You should elaborate more.

---

## Round 0.2 · Minor Revisions

The manuscript still needs attention. Please provide a high-resolution figure and, improve the discussion, as recommended by the Reviewer #1.

Reviewer 1 ·

Basic reporting

The author has made corrections from the first review. Now, the English language is better, and the unit to define the HIV viral load looks correct. Regarding literature references, the authors provided the data globally and in China that was previously unclear. The authors also provided all data to support their findings, including primers and technical details in material & methods. In general, results were relevant to supporting the hypothesis for a rapid and efficient HIV viral load detection kit.

Experimental design

No comment

Validity of the findings

I believe that the authors have provided supporting data to support their findings. It includes statements for the qPCR kit used for viral load detection, "DaAn Gene Co., Ltd. is CE-IVD/NMPA-approved ".

The primer probes used in this study are mentioned in Table 1, LOD in Table 2, Specificity in Table 3, and Specificity-sensitivity in Table 4. Figures clearly support their finding, including how this new method works in Figure 2. However, the image should be produced in high resolution to increase visibility.

Study limitations must be added in the discussion section. The conclusion in lines 283-284 can be improved, for example, by addressing the potential use of other areas outside China to improve test accessibility.

Additional comments

No comments

Reviewer 4 ·

Basic reporting

it seems the authors have addressed the reviewers comments, and implemented on the revised manuscript the suggestions

Experimental design

the authors have clarified the experimental design, and now it is suited for publication

Validity of the findings

they have improved the explanation of the study findings

Additional comments

the discussion has been improved describing the draw backs and advantages of the proposed diagnostic method in hand

---

## Round 0.3 · accepted · Accept

The authors have addressed all the reviewers' comments, and I am happy with the current version of the manuscript.